# Lamivudine (3TC), a Nucleoside Reverse Transcriptase Inhibitor, Prevents the Neuropathological Alterations Present in Mutant Tau Transgenic Mice

**DOI:** 10.3390/ijms241311144

**Published:** 2023-07-06

**Authors:** Laura Vallés-Saiz, Jesús Ávila, Félix Hernández

**Affiliations:** Centro de Biología Molecular “Severo Ochoa”, CSIC/UAM, Universidad Autónoma de Madrid, Cantoblanco, 28049 Madrid, Spain; laura.valles@cbm.csic.es (L.V.-S.); javila@cbm.csic.es (J.Á.)

**Keywords:** Alzheimer´s disease, lamivudine, transposons, tau

## Abstract

The dysregulation of transposable elements contributes to neurodegenerative disorders. Previous studies have reported an increase in retrotransposon transcription in *Drosophila* models as well as in human tauopathies. In this context, we tested the possible protective effects of a reverse transcriptase inhibitor, namely lamivudine (also known as 3TC), in P301S mice, an animal model of Alzheimer’s disease based on FTDP-17-tau overexpression. Transgenic P301S mice administered lamivudine through drinking water showed a decrease in the following histopathological marks typical of tauopathies: tau phosphorylation; inflammation; neuronal death; and hippocampal atrophy. Lamivudine treatment attenuated motor deficits (Rotarod test) and improved short-term memory (Y-maze test). To evaluate the role of tau in retrotransposition, we cotransfected HeLa cells with a plasmid containing a complete LINE-1 sequence and a neomycin reporter cassette designed for retrotransposition assays, and a plasmid with the tau sequence. LINE-1 insertion increased considerably in the cotransfection compared to the transfection without tau. In addition, lamivudine inhibited the insertion of LINE-1. Our data suggest that the progression of the tauopathy can be attenuated by the administration of lamivudine upon the first symptoms of neuropathology.

## 1. Introduction

Alzheimer´s disease (AD) is an age-dependent and progressive neurodegenerative disorder characterized by the accumulation of hyperphosphorylated tau in intracellular neurofibrillary tangles (NFTs) and extracellular amyloid plaques made of beta-amyloid peptide. In parallel, the central nervous system (CNS) is subjected to the loss of neurons. Most AD cases are sporadic, without a known underlying cause. The transposon aging theory hypothesizes that epigenetically silenced transposon elements (TEs) are reactivated as cellular defense and surveillance mechanisms begin to fail as a result of aging [1,2]. In *Drosophila*, the loss of heterochromatin with aging leads to activations of transposons [2]. The activation of TEs in the brain with aging and its relationship with functional neuronal deterioration was described for the first time in *Drosophila* [3]. Given that aging is the main risk factor in neurodegenerative diseases such as AD, the study of TEs in these conditions can shed light on these pathologies and reveal how to achieve healthy aging.

Half the mammalian genome comprises transposable elements [4]. Non-Long-Terminal Repeats (LTRs) are the major class of retrotransposons, and the most abundant is LINE-1 (L1). L1 expression and copy number variation in the human brain has been described previously [5]. L1-encoded proteins serve to reverse-transcribe LINE RNA and then insert the newly generated DNA copies into genomic DNA. L1 is composed of a 5′-UTR followed by two open reading frames (ORFs) and a 3′UTR with the polyadenylation signal. ORF1 codes for an RNA-binding protein (ORF1p), whereas ORF2 codes for ORF2p, which comprises endonuclease and reverse transcriptase activities [6,7]. It is now understood that transposition can provide an evolutionary advantage by driving genome expansion and evolutionary diversity. However, this process can cause DNA breaks and genomic instability, genetic polymorphism, and insertional mutations that can be detrimental to the organism [8]. Cellular systems that keep TEs under control decline with age, and the loss of this regulation contributes to age-related decline in tissue function [3,9,10,11,12]. Retrotransposon activation has been documented in the context of physiological aging of the *Drosophila* fat body and brain [3,10]. In the liver and muscle of aging mice, several retrotransposons are elevated at both the RNA and DNA levels, thereby suggesting that age-associated transposition also occurs in the somatic cells of mammals [9].

Recent studies have identified links between pathogenic forms of tau and TE-induced neurotoxicity. Thus, an increase in TE transcripts has been observed in a *Drosophila* model of mutant tau [13,14]. Furthermore, pathogenic tau seems to accelerate TE activation with age in the CNS of transgenic mice [15]. The deregulated expression of TEs in post-mortem brain tissues of AD patients has been also reported [16]. In human AD, the activation of retrotransposons is significantly associated with tau tangle burden and reduced cognitive performance [13]. Studies in fly models of tauopathy suggest that retrotransposon activation is caused by tau-induced heterochromatin breakdown, and retrotransposon silencing mediated by piwi-interacting RNA (piRNA) is a causal factor of neurodegeneration. These effects can be suppressed by antiretroviral therapy [14]. Thus, the activation of TEs could contribute to maintaining, amplifying, or even initiating neurodegeneration, and heterochromatin decompression could be a marker of TE activation and a common feature of neurodegeneration [11,17,18]. Nucleoside reverse transcriptase inhibitors (NRTIs) have been used in patients with human immunodeficiency virus (HIV) and, interestingly, antiretroviral therapy reduces the risk of neurocognitive disorders and dementias [19].

Here we report on the effects of treatment of the tauopathy mouse model P301S with lamivudine (also known as 3TC) in drinking water. 3TC, which is an NRTI, is an analog of cytidine. It inhibits reverse transcriptases and is used mainly in HIV therapy [20,21]. It shows an acceptable safety profile in long-term treatments in humans. 3TC suppresses tau-induced TE mobilization and tau-induced neurotoxicity in *Drosophila* [14]. However, it has not been studied in a mammalian model of tauopathy. Lamivudine rescues cognitive deficits in a mouse model of Down syndrome [22] and also improves cognitive decline in senescence-accelerated prone 8 (SAMP8) mice [23]. These observations thus point to the therapeutic potential of 3TC for the treatment of neurodegenerative diseases. We here extend these studies to validate 3TC for the treatment of AD patients. We observed a decrease in the histopathological marks typical of tauopathies, attenuated motor deficits, and an improvement in learning in 3TC-treated P301S mice. Furthermore, we demonstrate that tau increases the insertion of L1-retrotransposon in a cellular model, a phenomenon that is prevented by 3TC. These results support the notion that targeting retrotranscriptase with lamivudine might offer a potential therapeutic strategy for tauopathies.

## 2. Results

### 2.1. P301S Mice Show an Increase in Survival and an Improvement in Learning after 3TC Treatment

First, we treated the tauopathy mouse model P301S (also known as PS19 [24]) with 3TC, an NRTI that inhibits enzymes with reverse transcriptase activity. Figure 1A shows a schematic diagram of the experimental design used to explore whether 3TC treatment in drinking water at a dose of 100 mg/kg [23] for 3 months improved the life expectancy of these animals. As previously described, the P301S line showed motor deficits and paralysis, particularly focalized in the hindlimbs [24,25]. Paralysis in the hindlimbs worsened at around 10–12 months of age. However, mice treated with 3TC showed an improvement in this phenotype (Figure 1B). Thus, paralysis focalized in the hindlimbs was present only in 28% of 3TC-treated animals compared with 57% in untreated P301S counterparts.

3TC-treated wild-type (Wt) mice presented a similar life expectancy as untreated mice. In contrast, only 64% of untreated P301S mice survived at 11 months while 90% of those treated with 3TC in drinking water for 3 months were still alive at this time point (Figure 1C).

These findings indicated that 3TC treatment is sufficient to prolong the life expectancy of P301S mice. Given that these mice display motor deficits that are likely to lead to death, we studied paralysis with a rotarod test. Treated P301S mice outperformed untreated counterparts on this test (Figure 1D) (*p* < 0.05). To determine whether the 3TC treatment could delay the memory alterations present in P301S mice, we used the Y-maze test to assess short-term memory (Figure 1E). Significant differences in the time spent in the new arm were observed between Wt and untreated P301S mice (*p* < 0.015), but not between Wt and P301S mice treated with 3TC. Furthermore, differences in performance were detected between the P301S groups (*p* < 0.035), with 3TC-treated P301S mice showing a similar discrimination index to that of treated Wt mice. These results confirmed that treatment with 3TC delays the appearance of phenotypic and behavioral alterations detected in P301S mice.

### 2.2. Pharmacological 3TC Treatment Modulates P301S-Dependent Tau Pathology In Vivo

To determine whether tau hyperphosphorylation was altered in P301S mice treated with 3TC for 3 months, we compared tau phosphorylation in these mice (Figure 2). We used the antibody AT8, which can recognize tau protein phosphorylated at both serine 202 and threonine 205 [26]. An increase in AT8 immunostaining was evident in the cortex (Figure 2A,B) and also the CA1 region of the hippocampus (Figure 2C,D). Interestingly, two patterns of AT8 immunostaining were observed. Thus, we focused our analysis on those cells with an NFT-like or Type 1 phenotype (Appendix A) and on those with diffuse-tau phosphorylation or Type 2 phenotype (Appendix A). We found fewer diffuse phosphorylated tau cells in the hippocampus (*p* = 0.006) and cortex (*p* = 0.0004) of treated mice, while no differences were observed in NFT-like structures, the latter consistent with previous reports on chronic lithium administration to transgenic mice overexpressing mutant tau and GSK-3β [27]. Thus, when 3TC was administered to 8-month-old mice for three months, phosphorylated tau decreased, although tau aggregated in NFT-like structures was not affected.

### 2.3. Partial Reversal of Atrophy, Neuronal Death, and Gliosis in 3TC-Treated P301S Mice

Having demonstrated the beneficial effects of 3TC on the tauopathy mouse model, we proceeded to analyze their consequences. Differences in DAPI (nuclear marker) staining were observed between Wt and P301S untreated controls (Figure 3A). This was not detected in Wt and P301S mice treated with 3TC (Figure 3B), demonstrating atrophy of the hippocampus and mainly of the dentate gyrus. Moreover, in the lateral sagittal brain sections, we observed a substantial decrease in the volume of the lateral ventricle in 3TC-treated P301S mice compared with untreated counterparts, thereby suggesting that the neurodegenerative processes taking place in these mice were partially prevented. To further analyze the brain atrophy observed, we used brain sections to test whether ventricle atrophy occurred to the same extent in control and 3TC-treated P301S mice (Figure 3C). As compared to untreated P301S mice, 3TC-treated counterparts showed a 23.8% (*p* = 0.009) decrease in lateral ventricle size, reaching a size similar to that observed in the Wt animals (Figure 3D).

Brain atrophy and a concomitant increase in ventricle size often accompany neuronal loss and serve as a hallmark lesion for neurodegeneration. Cleaved caspase-3 contributes to neuropathology in P301S mice [28] and untreated P301S mice presented significant differences with respect to Caspase-3 staining in the cortex and hippocampal samples (Figure 4A–D). Again, differences among 3TC-treated mice were observed. In this regard, cortical (*p* = 0.0031) and hippocampal (*p* = 0.039) samples from treated P301S mice showed a significant reduction in the number of apoptotic cells. These data suggest that reverse transcriptase inhibition partially prevents cell death and concomitant brain atrophy.

Microgliosis and reactive astrogliosis often accompany neuronal loss. Thus, 3TC-treated Wt mice showed significant changes in microgliosis, as reflected by immunofluorescence performed with an antibody raised against CD68. CD68 expression decreased in the cortex of treated Wt mice compared to untreated Wt counterparts (Figure 5B, * *p* = 0.035), thereby demonstrating that lamivudine improved the microgliosis observed in aged Wt mice. Accordingly, the increase in microgliosis observed in the cortex (Figure 5A,B) and hippocampus (Figure 5D,E) of P301S mice was partially reversed in 3TC-treated mice, although significant differences were not observed. With respect to the reactive astrocytosis in P301S mice, as shown by immunofluorescence performed with an antibody raised against GFAP (Figure 5), a remarkable increase in GFAP-immunolabeling was observed in the cortex (Figure 5A,C) and CA1 region of the hippocampus (Figure 5D,F). Again, astrogliosis decreased, mainly in cortical samples, in 3TC-treated mice (*p* < 0.005; P301S vs. Wt mice).

Increased levels of the proinflammatory cytokine interleukin-1β (IL-1β) are an integral part of the local tissue reaction to CNS insult. Pro-inflammatory IL-1β in the untreated P301S cortex samples increased to 190.5 ± 7.5% (*p* < 0.001) of Wt controls. In contrast, 3TC-treated P301S mice did not show any significant difference compared with treated Wt mice (Figure 5G,H).

### 2.4. Histone-3 Trimethylation Levels and LINE-1 Insertion in Genomic DNA

H3K9me3 at lysine 9 is a marker for heterochromatin, and loss of heterochromatin with aging lifts the silencing of genes repressed, including TEs [2,29,30,31]. To test whether P301S mice show a decrease in histone methylation, as has been described previously in aging in the hippocampal dentate gyrus (DG) [31], we examined the levels of H3K9me3 in these animals. H3K9me3 levels in the granular cells of the DG of 11-month-old P301S mice decreased compared with Wt mice (*p* < 0.001) (Appendix A). However, when the same experiment was performed with 3TC-treated mice, these differences were not observed. The same analysis was carried out in the cortex and CA1 hippocampal area, although no significant differences were observed (Appendix A), thereby suggesting that the changes in the expression of this marker are region-dependent. We next addressed whether L1 retrotransposon insertion in genomic DNA is elevated in the brains of P301S mice, as retrotransposition events increase the total number of DNA copies of a given transposon. To test this hypothesis, we analyzed L1 insertion, as levels of this retrotransposon have been reported to be increased in AD [13]. L1 retrotransposons did not differ significantly between wild-type and P301S mice (*p* = 0.16). Interestingly, an increase in L1 DNA copy number was observed in Tg4510 mice that overexpress Tau-P301L [15]. 3TC treatment of P301S mice reduced transposition activity to background levels, similar to what was observed in Wt mice (Figure 6A). However, the differences between 3TC-untreated and untreated P301S mice were not significant (*p* = 0.23).

### 2.5. Tau Promotes LINE-1 Insertion In Vitro

3TC treatment attenuated motor deficits in the P301S mice and the progression of the tauopathy. Therefore, we analyzed in greater detail whether the overexpression of tau protein interferes with the insertion of L1 in the genome of HeLa cells. To this end, the plasmid pBS-L1PA8CHmneo [32] with complete L1 and a neomycin reporter cassette designed for retrotransposition assays was used. To study whether tau is involved in the insertion of the transposons, cotransfections were made with the plasmid pSGT42 [33], which expresses the longest human tau isoform present in the CNS. In this experiment, we stained colonies with crystal violet to determine the number that had inserted L1 and were therefore resistant to neomycin. Insertion increased considerably and clearly in the cotransfection experiment with Tau42 compared to the cotransfection with the empty plasmid pSG5 (Figure 6B,C). Thus, there were almost two times more colonies in the presence of tau than in its absence (*p* = 0.012). In parallel, when the 3TC treatment was given, the cytidine analog prevented L1 insertion completely.

## 3. Discussion

Here we sought to explore whether the retrotranscriptase inhibitor 3TC prevents tau pathology in a tauopathy mouse model based on the overexpression of human tau with a mutation present in patients with frontotemporal dementia with parkinsonism-17 (FTDP-17) and whether tau alters retrotransposon levels. Our study demonstrates that 3TC treatment halts the progression of tau pathology with regard to the memory deficits and motor deficits characteristic of this model and that tau overexpression increases L1 transposition in cellular cultures.

One of the main histopathological marks of AD is the presence of hyperphosphorylated tau. However, recent studies demonstrate that other markers such as heterochromatin relaxation [34] and genomic instability [35,36] are also markers of neurodegeneration. Furthermore, in contrast to the initial notion that the neuronal genome is a fixed unit, several lines of evidence have revealed that retrotransposon expression is increased in neurodegeneration, possibly causing retrotransposon-mediated gene upregulation and insertional mutations [35].

Here, we first showed that 3TC treatment improved the memory and motor deficits present in P301S mice and prolonged lifespan. The treatment was initiated at the age of 8 months, when the first signs of tau pathology are observed [25], and finished 3 months later when full tau pathology occurs. Of note, the 3TC treatment was sufficient to cause a decrease in the number of diffuse-type AT8-positive structures in the cortex and the hippocampus. In good agreement, 3TC-treated P301S mice presented less brain atrophy and neuronal death in these two regions of the brain than untreated mice. This protective effect was accompanied by an improvement in gliosis and a reduction in IL1β levels. We have used a higher dose in our animal model than that used in humans. Therefore, the 3TC dose administered here (100 mg/kg [23]) is much higher than the 300 mg daily dose administered to human HIV carriers. However, in the case of mice, the drug is administered in the drinking water and, therefore, its administration is carried out throughout the day. Clearly, further studies are needed to find out whether a higher dose of 3TC is required to inhibit transposon reverse transcriptase than to inhibit HIV virus reverse transcriptase and whether there are differences in 3TC toxicity.

dPCR (digital PCR) has previously revealed that L1 levels are increased in P301S mice [15], an observation that we reproduced by qPCR, although without a significant level. Next, we analyzed in Hela cells the effect of tau protein overexpression on L1 insertion. We confirmed that tau overexpression is sufficient to increase L1 insertion and that 3TC inhibits this integration. However, it should be noted that an increase in transposon insertion does not necessarily mean an increase in L1 copy number. In fact, most L1 insertions are not full-length insertions, but rather 5′-truncated forms (for a review [37]).

L1 integration in P301S mice may occur because these animals show chromatin dysregulation. Epigenetic modifications like H3K9me3 may regulate human aging [38]. We found that the expression of P301S-tau results in a decrease in H3K9me3 in granule cells in the DG. Interestingly, alterations in H3K9me3 were prevented by a 3-month treatment with 3TC. It is unclear how mutant tau could facilitate H3K9me3 demethylation, although several mechanisms can be put forward. Thus, the direct binding of tau to DNA occurs [39] (for a review see [40]). Interestingly, a loss of heterochromatin due to tau depletion has been described, something that may happen in P301S mice as a loss of function of mutant tau [38]. In addition, a correlation was reported between tau pathology and chromatin relaxation [13,34]. The colocalization of tau with heterochromatin points to a potential role of tau in the organization and protection of chromatin during the aging process [41]. The role of tau in age-associated chromatin alterations should be further considered given that AD and aging exhibit increased levels of retrotransposons and open chromatin due to the loss of heterochromatin [13,14]. Thus, a possible explanation of how mutant tau could facilitate H3K9me3 demethylation is that it alters nuclear morphology and metabolism. Tau regulates the nuclear pore complex [40,42,43], and the expression of full-length tau or truncated tau in neuroblastoma cells leads to nuclear envelope invaginations [44]. Regarding the latter, they are filled by tau nuclear rods in neuropathologies like Huntington’s disease [45] and in P301S mice [46], possibly deforming the nuclear membrane [47]. Likewise, the tau mutant P301L-transgenic mouse shows disruption of the nuclear lamina and, consequently, disruption of nucleocytoplasmic transport [42].

Another explanation of how pathogenic tau alters transposon nucleo/cytoplasmatic traffic is provided by the capacity of tau to directly interact with nucleoporins of the nuclear pore complex and affect their structural and functional integrity [42]. However, given the capacity of piRNAs to regulate transposon levels, the involvement of piRNA should also be studied in a similar way to the mechanisms of piRNA-mediated transcriptional silencing in mouse germ lines. In addition to reports that pathogenic tau increases transposon deregulation in *Drosophila* [14], piRNA levels in this model decrease until exhaustion, which promotes neuronal death through TE dysregulation in tauopathies [11].

One of the main conclusions of our work is that 3TC prevents tau pathology but not when NFTs have been already formed suggesting that 3TC should be given to Alzheimer’s patients in the prodromal stage of the disease before the formation of any tau tangles. However, there is no direct evidence showing that aggregated tau is toxic for neurons. In fact, the formation of aggregates has been proposed to be a defense mechanism in other neurodegenerative diseases (i.e., Huntington’s disease) [48]. Consistent with this, a learning deficit in a mouse model of FTDP-17 correlates with hyperphosphorylated tau but not with the presence of aggregated tau in NFTs [49]. Thus, we propose that 3TC could also be effective even at advanced stages of the disease. However, the transgenic model used in this study is based on the overexpression of human tau protein with a mutation present in patients with FTDP-17. It will be interesting to extend these studies to analyze the effect that this drug has on a model of Alzheimer’s disease that includes the presence of senile plaques and NFTs.

In summary, to the best of our knowledge, this is the first study to demonstrate that the administration of lamivudine (3TC), an NRTI drug used to treat HIV and with well-documented effects in humans [20,21], prevents tau-associated neurodegeneration. Given our findings, 3TC emerges as a potential therapeutic approach for neurodegenerative tauopathies. Our data also support studies based on novel retrotranscriptase inhibitors as new pharmacological treatments for this kind of neurodegenerative disorder. A Pilot Study to Investigate the Safety and Feasibility of AntiRetroviral Therapy for Alzheimer’s Disease (ART-AD) (https://clinicaltrials.gov/ct2/show/results/NCT04552795 (accessed on 4 May 2023) has been proposed based on previous studies, although no results have been reported yet. In this regard, our results in a mammalian model support this approach and provide baseline data for future trials.

## 4. Material and Methods

Animals and Experimental design: B6;C3-Tg(Prnp-MAPT*P301S)PS19Vle/J [24] mice from the Jackson Laboratory (stock 008169, Bar Harbor, ME, USA) were used. Littermate wild-type (Wt) mice were used as controls. Mice were housed in a specific pathogen-free colony facility, in accordance with European and local animal care protocols. Animal experiments received the approval of the CBMSO’s (AECC-CBMSO-13/172) and national (PROEX 102.0/21) Ethics Committees. Mice were fully anesthetized by an intraperitoneal injection of pentobarbital (Dolethal, Vetoquinol, 60 mg/mL) and transcardially perfused with 0.9% saline. Four groups were used: Wt and P301S not given 3TC treatment n = 13–15 and those given 3TC treatment n = 13–15. 3TC (lamivudine) was administered in drinking water at a dose of 100 mg/kg [23] for 3 months.

Tissue Processing: The mice were fully anesthetized with an intraperitoneal pentobarbital injection (Dolethal, 60 mg/kg body weight) and transcardially perfused with 0.9% saline. The brains were removed and postfixed overnight in 4% paraformaldehyde. For immunofluorescence experiments, 50-μm-thick sagittal brain sections were obtained on a Leica VT1200S vibratome. The animals used for biochemical analysis were fully anesthetized with an intraperitoneal pentobarbital injection and transcardially perfused with saline only. The brains were removed and the hippocampi were quickly dissected on ice and frozen in liquid nitrogen.

Rotarod Test: An Ugo Basile was the equipment used as described previously [25]. Mice had two training days. On the first day, the mice were subjected to a speed of 4 rpm with constant acceleration for 1 min four times. On the second day, the mice were subjected to an acceleration from 4 to 8 rpm over 2 min four times. On the test day, the time the mice stayed on the apparatus without falling was timed with acceleration from 4 rpm to 40 rpm, for 5 min four times. The mean of the latencies of 4 trials per mouse was calculated. On each day, series were separated by 30 min, 1 h, and 30 min.

Y-maze test: Working memory and exploratory activity were measured using a Y-maze apparatus (three equal arms). Each mouse was placed at the start of arm A. In the trial, one arm (B or C) was closed for 15 min (familiarization phase). Two hours later, with all the arms open, the time each mouse spent in the new arm was evaluated over 10 min (test phase). Mice were automatically tracked using a computerized video system. Results are expressed using a discrimination index (DI) calculated as follows: time in the novel arm divided by the sum of both.

Western blot: Brain samples were homogenized in 20 mM HEPES pH 7.4, 100 mM NaCl, 50 mM NaF, 5 mM EDTA, and 1% Triton X-100 supplemented with protease and phosphatase inhibitors. The protein concentration of each homogenate was determined by the Bradford method using the BCA test (Thermo Fisher, Waltham, MA, USA). Finally, the SDS-PAGE buffer (250 mM Tris pH 6.8, 4% SDS, 10% glycerol; 2% β-mercaptoethanol and 0.0006% bromophenol blue) was added to the protein extracts obtained. The extracts were boiled in a thermoblock at 100 °C for 5 min. Thirty micrograms of protein per well was loaded from each sample. The proteins were separated on 10% acrylamide/bisacrylamide gels in the presence of SDS at 100 mV for approximately 1 h. Those present in the gel were transferred to nitrocellulose membranes (Schleicher and Schuell, Keene, NH, USA) at 150 mA for 50 min, using the Bio-Rad Mini-Protean system. Subsequently, the membranes were blocked using 5% milk powder in 0.1% Tween PBS for 1 h. The membranes were then washed twice with 0.1% PBS Tween-20 (*v*/*v*) under stirring for 10 min. They were then stained with Ponceau dye (Ponceau 0.3% in TCA 3%) to check transfer efficiency. Finally, they were incubated with the appropriate primary antibody overnight at 4 °C: anti-IL1β (goat, 1:1000, R&D Systems, MN, Minnesota, USA), anti-phosphorylated tau (AT8, mouse, 1:1000, Innogenetics, Ghent, Belgium), and total-tau (Tau-5, mouse 1:1000, Abcam, Cambridge, UK) and anti-vinculin (rabbit, 1:5000, Abcam). Protein expression was detected using the secondary antibody (1:1000), which was incubated for 1 h at room temperature. After performing three 10-min washes in the wash solution, the immunoreactive proteins were detected using an Enhanced Chemiluminescence Detection System (Amersham, Amersham, UK).

Immunohistochemistry and immunofluorescence: The samples used for immunolabeling were fixed with 4% PFA in 0.1N phosphate buffer (PB) overnight at 4 °C. They were then washed three times in PB and placed in a 10% sucrose/4% agarose matrix. Sagittal brain sections were subjected to a floating immunolabeling process.

Immunohistochemistry of 3,3′-diaminobenzidine (DAB); the VECTASTAIN Elite AB kit (Vector Laboratories, Newark, CA, USA) was used. The sections were washed with Phosphate-buffered saline (PBS) to remove the cryoprotective solution. Subsequently, they were immersed in H_2_O_2_ to 0.33% in PBS for 45 min to block the activity of endogenous peroxidase. After three washes, the sections were placed in a blocking solution (PBS with 0.5% bovine fetal serum, 0.3% Triton X-100 and 1% BSA) for 1 h and incubated overnight at 4 °C with the corresponding primary antibody diluted in blocking solution: Anti-Caspase 3 (rabbit, 1:100, Cell Signaling, Danvers, MA, USA). The next day, the sections were washed three times for 10 min with PBS. They were then incubated first with the biotinylated secondary antibody for 1 h and then with the avidin-biotin-peroxidase from the kit for 1 h. The developing reaction was performed using DAB for approximately 10 min. Finally, the sections were placed in slides using FluorSave (Merck Millipore, Darmstadt, Alemania) as a mounting medium. Images were taken using an upright microscope Axioskop2 plus (Zeiss, Jena, Germany) coupled to a DMC6200 camera (Leica, Wetzlar, Germany). For immunofluorescence, after washing the sections three times with PB 0.1N, they were incubated with the following primary antibodies in the blocking solution (1% BSA and 1% Triton X-100 in 0.1 N PB) at 4 °C for 24 h: Anti-H3K9me3 (rabbit, 1:1500, Abcam); Anti-AT8 (mouse, 1:500, Innogenetics); Anti-CD68 (Rat, 1:500, Abcam); and Anti-GFAP (chicken, 1:2000, Abcam). The next day, the sections were washed five times with the same blocking solution and then incubated with the corresponding secondary antibodies conjugated with Alexa fluorophores at 4 °C with gentle agitation for 24 h (1:1000, Molecular Probes, Eugene, OR, United States). Finally, the sections were washed three times with PB an then with DAPI diluted 1:5000 for 10 min, and another three additional washes were performed with PB. They were then placed on slides using FluorSave.

qPCR: The relative expression of L1 was determined by qPCR with gDNA. gDNA was isolated with the Maxwell 16 DNA Tissue DNA Purification Kit (AS1030, Promega, Alcobendas, Spain) following the manufacturer’s protocol. qPCR was performed using SYBR Green reagent (Applied Biosystems, Waltham, MA, USA) and ABI PRISM 7900HT SDS (Applied Biosystems, Waltham, MA, United States) equipment. The reaction per well was 10 μL and contained 0.5 μL of gDNA template per sample to bring it to 25 ng/well and bringing this volume to 4 μL with H_2_O, in addition to 5 μL of the SYBR Green PCR mix and 1 μL per oligonucleotide pair at 5 μM. The primers used were L1 forward primer: 5′-TAAAACGAACAGCACCTTGGG-3′ and L1 reverse primer: 5′-ATTGCATCTCCTTCTTGCTGC-3′. qPCR amplification of genes was performed for 40 cycles of 95 °C for 1 s and 60 °C for 20 s. No amplification from the no template control (NTC) was observed for the genes of interest. ValidPrime kit (PN A106510, Tataa Biocenter, Göteborg, Sweden) mouse was used as a normalizer. Each primer pair showed a single sharp peak, thereby indicating that the primers amplified only one specific PCR product. Three technical replicates per gene were used.

Confocal microscopy: Confocal stacks of images were obtained with a Confocal Spinning Disk SpinSR10 attached to an IX83 inverted microscope (Olympus, Tokyo, Japan). The images were taken in three regions of the brain, namely the DG, CA1, and cortex. In all cases, the photo was taken at the same localization. To this end, each structure was first identified by an overview of a 10× dry objective in the DAPI channel. Next, at least three slides per mouse were used. The acquisition settings (laser intensity, gain, and background subtraction) were kept constant for all images.

Image Acquisition and cell counting: Each image acquired in medial sagittal brain sections was taken between 2.04 mm (Figure 118 of the atlas of Paxinos and Franklin [50]) and 0.60 mm (Figure 106 of the same atlas) with respect to the midline. Sagittal brain sections (3 slices per mouse and at least 13–15 mice per group) were used for cortical and hippocampal cleaved Caspase-3+ cell counts. Immunohistochemistry techniques were used to identify the cleaved Caspase-3+ cells, counting the number of immunostained reactive cell bodies present in each slice. To count caspase-positive cells, we used the color deconvolution tool in Fiji and selected the DAB H vector. This tool generates three windows with different colors. We used these three windows to visually count the positive cells and distinguish them from small DAB precipitates, among other artefacts. The criterion for the size of the positive signal for DAB to consider that it was a caspase-positive cell was always the same. The lateral ventricle area was determined by measuring the ventricle area in at least three serial lateral sagittal sections between 3.4 mm (Figure 130 of the atlas of Paxinos and Franklin [50]) and 3.0 mm (Figure 126 of the same atlas) with respect to the midline (harvested every 50 μm) per mouse in at least 13–15 mice per genotype.

To quantify the number of hippocampal AT8+ cells, stacks were obtained under 40× oil-immersion objective and the Cell counter plugin of Fiji was used. Two cell types were identified, namely Type 1 (Tangle-like), these being cells with high levels of tau aggregates, and Type 2 (Diffuse tau), these being AT8^+^ cells with a smoother signal (Appendix A). The CA1 area of the hippocampus was identified by staining using DAPI dye.

The percentage of the area occupied by CD68 and GFAP in the DG, CA1, and cortex stacks was obtained under 20× oil-immersion objective. The invariant threshold (Triangle algorithm) was then set in Fiji for both channels. Next, the percentage of the area occupied was measured. Confocal images were acquired from three separate medial sagittal brain slices per mouse on at least 13–15 mice per condition.

To quantify the H3K9me3 fluorescence intensity in the DG, all stacks were obtained under 20× oil-immersion objective. The invariant threshold (Otsu algorithm) was then set in Fiji for the H3K9me3 channel. The DG area was drawn and that above the threshold was measured in Fiji. Confocal images were acquired from three separate sagittal brain slices per mouse on at least 13–15 mice per condition.

Evaluation of the retrotransposition of L1 insertion in HeLa cells: The following constructs were used: plasmid pBS-L1PA8CHmneo (Plasmid #69608, Addgene, Watertown, MA, USA) designed to select the colonies that have undergone retrotransposition expresses the codon-optimized ORF1 and ORF2 of the consensus sequence of L1PA8 and the construct is tagged with the neomycin indicator cassette designed for retrotansposition assays as if retrotransposition does not occur, the resulting cell will not be resistant to neomycin [32], and pSG-T42R, which expresses the codon tau [33]. The empty vector pSG5 was used as a control.

HeLa cells (CRM-CCL-2^TM^, ATCC) were seeded in M6 at a density of 150,000 cells/well in 10%DMEM. We added 3TC (Sigma, St. Louis, MO, USA) at 50 μM to control wells at the beginning of the experiment. Transient transfections were performed the following day using Lipofectamine LTC reagent (Invitrogen, Waltham, MA, USA) following the manufacturer’s protocol. The cells were transfected using 1 μg of the constructs of interest. After 48 h, the cells were treated with the appropriate selection media containing Neomycin/G648 400 μg/mL (Gibco, Thermo Fisher, Waltham, MA, USA). After 14 days, cells were fixed and stained for 30 min with crystal violet (0.2% crystal violet in 5% acetic acid and 2.5% isopropanol). We added 3TC at the time of media changes for a period of 7 days.

Statistical analyses: Data are expressed as mean SEM. Statistical analyses were performed using Prism 9 (GraphPad Software, La Jolla, CA, USA). The D’Agostino and Pearson omnibus normality test was used to check the normality of sample distribution. For the comparison of means between more than two experimental groups with two variables, data were analyzed by a two-way ANOVA test. For comparisons of means between two groups, two-tailed unpaired *t*-tests were performed. A 95% confidence interval was used for statistical comparisons. It was considered statistically significant * *p* value < 0.05; ** *p* value < 0.01 and *** *p* value < 0.001.

## Figures and Tables

**Figure 1 ijms-24-11144-f001:**
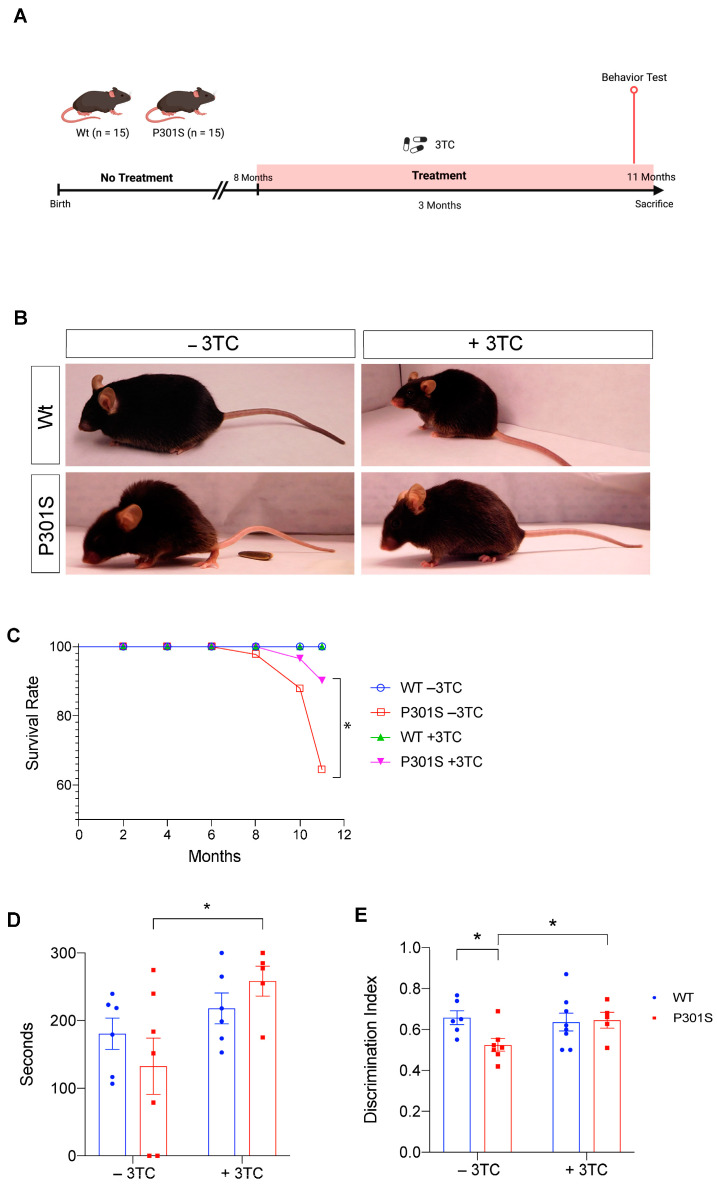
3TC treatment promotes an increase in survival and an improvement in learning in P301S mice. (**A**) Schematic diagram of the experimental design of 3TC treatment in drinking water at a dose of 100 mg/kg for 3 months. (**B**) Representative posture of 11-month-old P301S mice showing a hunched back and limb retraction, compared to age-matched wild-type animals. Effects of the 3TC treatment on the phenotype are also shown. (**C**) Kaplan–Meier curves for cumulative survival show statistically significant differences between 3TC-treated and untreated P301S mice (n = 13–15; *p* < 0.047). (**D**) Rotarod test; mean latency to fall from the rod of P301S mice is shown (seconds to fall). (**E**) Y-maze test; discrimination index between arms. * *p* < 0.05 Student’s *t*-test and long-rank test. +/−3TC: with or without lamivudine treatment in the drinking water (n = 5–7). Diagram created with BioRender.com.

**Figure 2 ijms-24-11144-f002:**
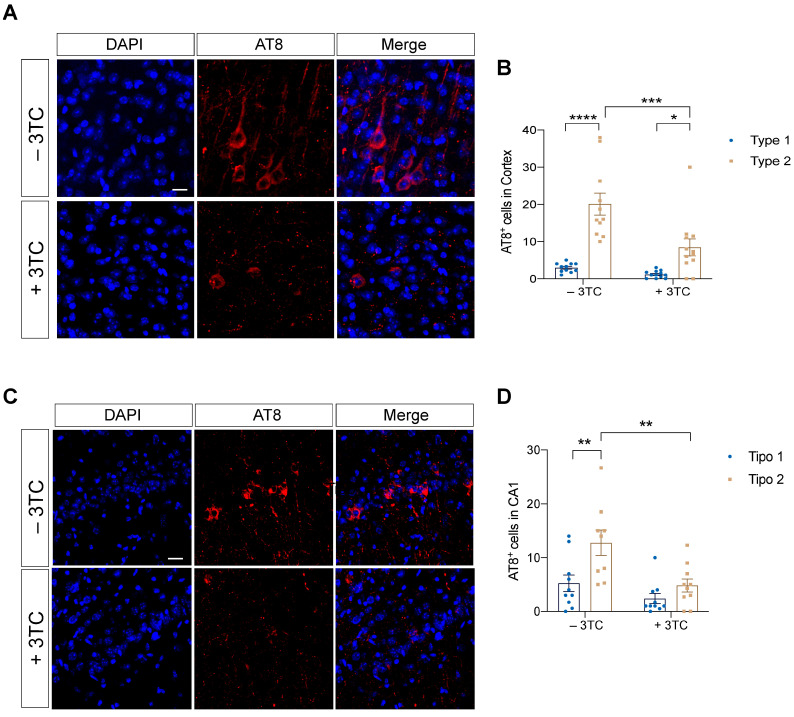
Decrease of the AT8 marker in the P301S mice treated with 3TC. Representative immunofluorescence images of AT8 labeling (red) in the cortex (**A**) and quantification of the same (**B**). Representative image of AT8 labeling in CA1 (**C**) and its quantification (**D**). DAPI staining of nuclei is shown in blue. The graphs show the mean ± SEM (n = 13–15 per condition). * *p* < 0.05; ** *p* < 0.01; *** *p* < 0.001; **** *p* < 0.0001 using two-way ANOVA followed by Student’s *t*-test for comparations. Scale bar, 20 μm.

**Figure 3 ijms-24-11144-f003:**
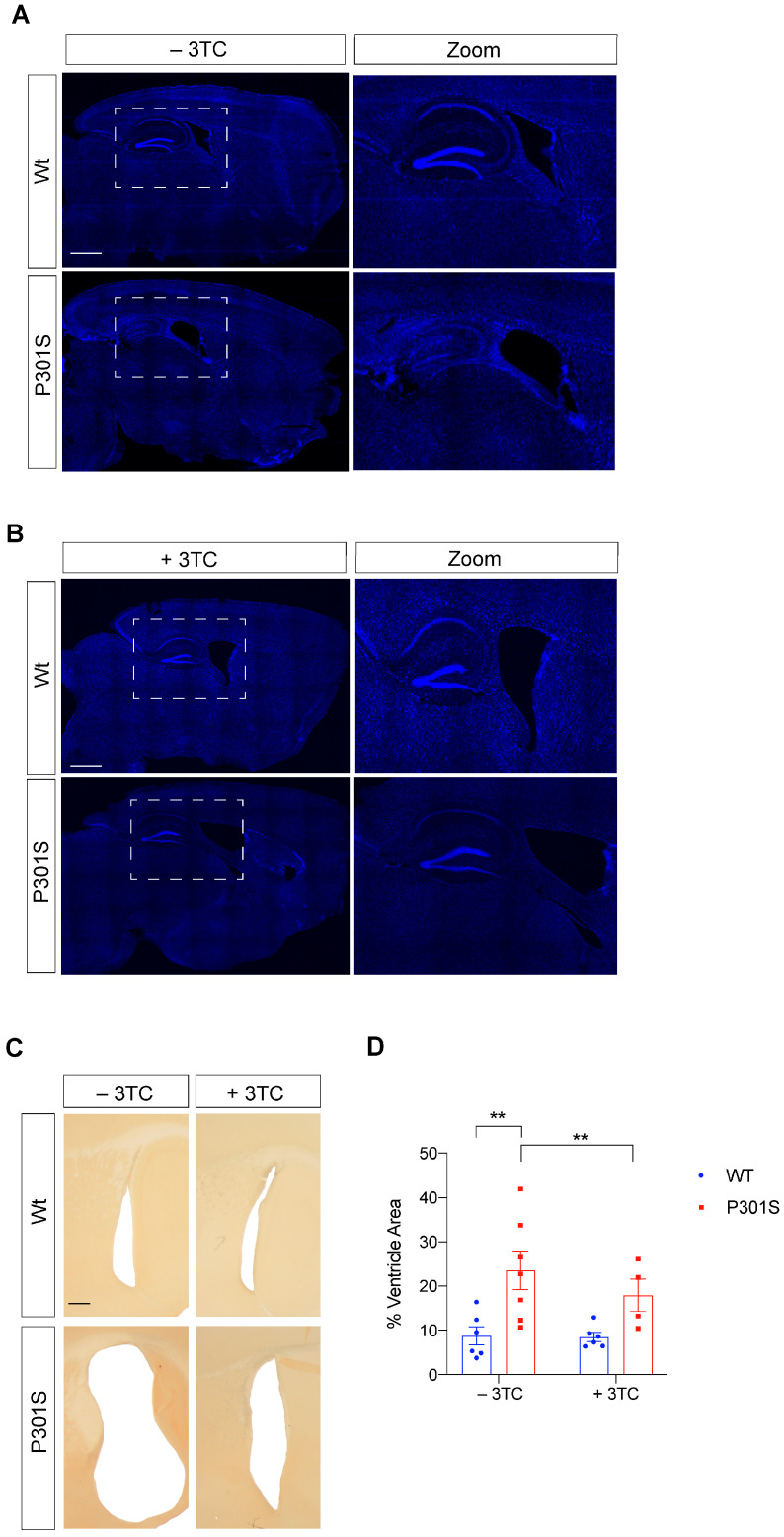
Lateral ventricular atrophy present in P301S mice decreased upon 3TC treatment. Representative images of the whole hippocampus. Cell nuclei of untreated mice (**A**) or 3TC-treated (**B**) mice were labeled with DAPI (blue). Scale bar, 1000 μm. High-power magnifications are shown on the right. (**C**) Representative lateral sagittal sections of the lateral ventricle (**D**) and quantification of lateral ventricular atrophy. Scale bar, 25 μm. The graph shows the mean value± SEM (n = 7 per condition). ** *p* < 0.01 using ANOVA followed by Student’s *t*-test for comparations.

**Figure 4 ijms-24-11144-f004:**
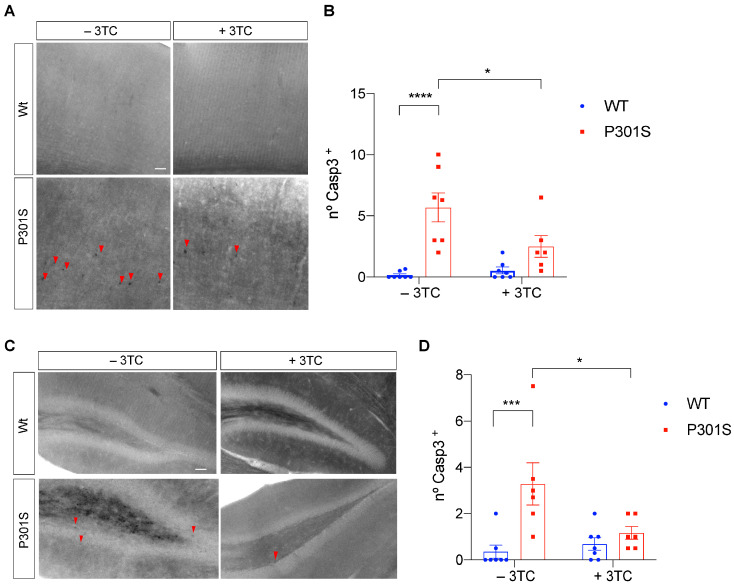
Reduction of Caspase-3+ cells in P301S mice treated with 3TC. Representative sagittal sections and quantification of the cortex (**A**,**B**) and hippocampus (**C**,**D**) of wild-type and P301S mice treated with 3TC. Slices were stained with Cleaved Caspase-3 antibody. The graphs (number of Caspase-3+ cells per section) show the mean value ± SEM (n = 13–15 per condition). * *p* < 0.05; *** *p* < 0.001; **** *p* < 0.0001 using ANOVA followed by Student’s *t*-test for comparisons. Arrowheads indicate immunostained reactive cell bodies. Scale bar, 20 μm.

**Figure 5 ijms-24-11144-f005:**
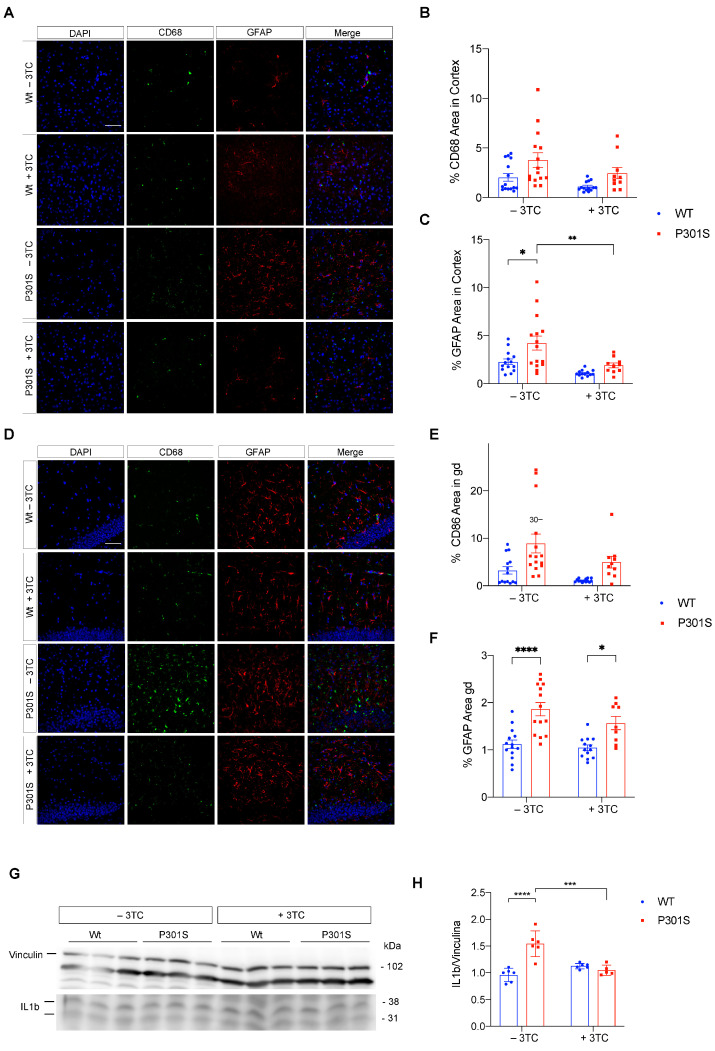
Gliosis and IL1β present in P301S mice treated with 3TC. Representative images of CD68 (green) and GFAP (red) labeling by immunofluorescence in the cortex (**A**) and its quantification (**B**,**C**). (**D**) Representative images of CD68 and GFAP labeling in CA1 and its quantification (**E**,**F**) (n = 13–15 per condition). ). DAPI staining of nuclei is shown in blue. (**G**) Representative immunoblot and (**H**) quantification of IL1 β in cortical homogenates for each of the four groups of mice analyzed. Levels of vinculin were used as loading control for normalization purposes. The arrows shown on the left are the bands quantified as described in Materials and Methods. The graphs show the mean ± SEM (n = 6–7 per condition). * *p* < 0.05; ** *p* < 0.01; *** *p* < 0.001; **** *p* < 0.0001 using two-way ANOVA followed by Student’s *t*-test for comparations. Scale bar, 20 μm.

**Figure 6 ijms-24-11144-f006:**
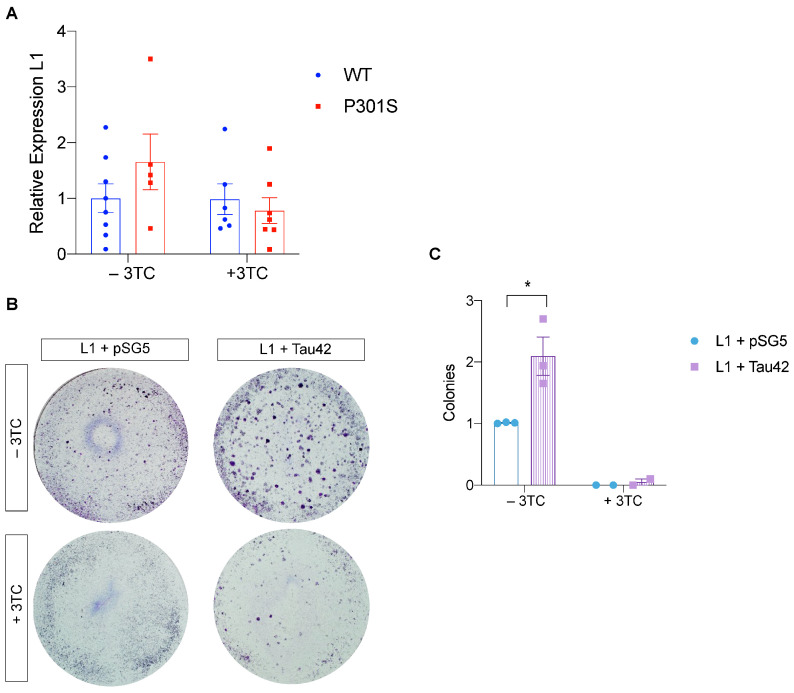
Increased insertion of L1 in the presence of tau in HeLa cells. (**A**) Variation of LINE-1 copies present in genomic DNA according to genotype and treatment (n = 6–7 per condition). DNA copy number is normalized to the average of untreated wild-type mice. (**B**) Crystal violet colony staining in pBS-L1PA8CHmneo (pL1) + pSG5 and pL1 + pSG-Tau42 cotransfections in HeLa colonies resistant to neomycin, in the presence or absence of the inhibitor 3TC (50 μM). (**C**) Quantification of the number of colonies normalized by assigning a value of 1 to the total number of colonies in the L1-pSG condition for each experiment (n = 3). The graphs show the mean value ± SEM. * *p* < 0.05 (n = 3 independent experiments).

## Data Availability

The datasets generated during the current study are available from the corresponding author upon reasonable request.

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
