# Peer review of "Lamivudine (3TC), a Nucleoside Reverse Transcriptase Inhibitor, Prevents the Neuropathological Alterations Present in Mutant Tau Transgenic Mice"

_ijms, 2023, doi:10.3390/ijms241311144_

Round 1
Reviewer 1 Report
Valles-Saiz et al. administered Lamivudine to P301S mice and observed its effects on motor impairment, cognitive dysfunction, and brain pathology. The results, demonstrating a reduction in impairments due to Lamivudine administration, are intriguing. To improve the quality of the paper, the following points should be addressed:
In Figures 1D and 1E, it appears that no multiple-group comparisons were conducted. It is necessary to specify the comparison groups. Additionally, in Figure 1D, there is a significant variation in the P301S-3TC group. While there are two individuals with Rotaload at 0 seconds, there are also two individuals with longer Rotaload values compared to the WT group. Could this variation be attributed to differences in the severity of paralysis?
It appears that there are experiments involving multiple-group comparisons and experiments comparing only two groups, but it is unclear which components were compared in each graph.
Was the Maze test conducted on mice with mild degrees of paralysis?
Regarding the AT8 staining results: The authors claim the presence of two types of staining patterns, NFT-like and diffuse. While only fluorescent immunostaining is presented, it would be beneficial to include immunohistochemical bright-field images of both weak and strong magnifications.
Regarding hippocampal atrophy: The extent of atrophy is difficult to discern, so it would be desirable to include H&E-stained images.
Figure 5: The rectangles in panels A and D appear misaligned, requiring readjustment.
Figure 6: In the graph of panel C, the Y-axis indicates colony numbers with a maximum of three. However, based on the image in panel B, it appears that there are more colonies present.
In the sentence starting from line 273, the authors claim, "Of note, the 3TC treatment was sufficient to cause a decrease in the number of neurons with hyperphosphorylated tau in the cortex and the hippocampus." However, since NFT-like inclusions did not decrease, it should be stated that the number of diffuse-type AT8-positive structures decreased.
Author Response
Valles-Saiz et al. administered Lamivudine to P301S mice and observed its effects on motor impairment, cognitive dysfunction, and brain pathology. The results, demonstrating a reduction in impairments due to Lamivudine administration, are intriguing. To improve the quality of the paper, the following points should be addressed:
In Figures 1D and 1E, it appears that no multiple-group comparisons were conducted. It is necessary to specify the comparison groups. Additionally, in Figure 1D, there is a significant variation in the P301S-3TC group. While there are two individuals with Rotaload at 0 seconds, there are also two individuals with longer Rotaload values compared to the WT group. Could this variation be attributed to differences in the severity of paralysis?
The group comparison was conducted pairwise, comparing two groups at a time. The P301S transgenic mice exhibit significant variation at the same age. This makes it so that while some P301S-3TC mice show similar phenotypic characteristics to WT mice, there are also mice in the early stages of paralysis. The Rotarod test is used to assess motor deficits, and it is important to consider the potential impact of weight on the results. For bioethical reasons, we sacrificed mice before the scheduled date of sacrifice (before the behavioral test too) if they exhibited a severe phenotype. Despite all of this, of course, the differences could be attributed to differences in the severity of paralysis.
It appears that there are experiments involving multiple-group comparisons and experiments comparing only two groups, but it is unclear which components were compared in each graph.
For the comparison of means, data were analyzed by a two-way ANOVA test. For comparisons of means between two groups, two-tailed unpaired t-tests were performed. A 95% confidence interval was used for statistical comparisons. It was considered statistically significant *p value<0.05; **p value<0.01 and ***p value <0.001. Only those comparisons with significant p values are shown.
Was the Maze test conducted on mice with mild degrees of paralysis?
There were animals with early stages of paralysis, but none of them had a high level of paralysis. The Rotarod test and Y maze test are very different in terms of the physical condition required from the animals. While the Rotarod test assesses motor coordination and locomotor function, requiring a greater physical effort from the mice, the Y maze test evaluates short-term memory and does not require significant physical exertion. This is why some animals that scored 0 in the Rotarod test performed average in the Y maze test.
Regarding the AT8 staining results: The authors claim the presence of two types of staining patterns, NFT-like and diffuse. While only fluorescent immunostaining is presented, it would be beneficial to include immunohistochemical bright-field images of both weak and strong magnifications.
Supplementary Figure 1 shows representative AT8+ Type 1 or "Tangle-like" neuron and Type 2 neuron with a diffuse pattern analysed in this study. We show below, for the interest of the reviewer, immunohistochemistry (DAB immunoprecipitation) with the same antibody where it can be seen both labeling patterns in the cortex of P301S mice.

Regarding hippocampal atrophy: The extent of atrophy is difficult to discern, so it would be desirable to include H&E-stained images.
Figure 3 presents a representative image of hippocampal atrophy present in untreated P301S mice. DAPI staining images are shown in Figures 3A and B. In our opinion, the measurement of such atrophy should be identical whether sections are stained with DAPI or H&E.
Figure 5: The rectangles in panels A and D appear misaligned, requiring readjustment.
That has been done.
Figure 6: In the graph of panel C, the Y-axis indicates colony numbers with a maximum of three. However, based on the image in panel B, it appears that there are more colonies present.
We do agree with the reviewer. We conducted three independent experiments and we normalized the data assigning a value of 1 to the total number of colonies in the L1-pSG condition for each experiment. We have changed figure caption 6 as follows:
Figure 6. Increased insertion of L1 in the presence of tau in HeLa cells. (A) Variation of LINE-1 copies present in genomic DNA according to genotype and treatment (n= 6–7 per condition). DNA copy number is normalized to the average of untreated wild-type mice. (B) Crystal violet colony staining in pBS-L1PA8CHmneo (pL1) + pSG5 and pL1 + pSG-Tau42 cotransfections in HeLa colonies resistant to neomycin, in the presence or absence of the inhibitor 3TC (50 μM). (C) Quantification of the number of colonies normalized by assigning a value of 1 to the total number of colonies in the L1-pSG condition for each experiment (n=3). The graphs show the mean value± SEM. *p<0.05 (n=3 independent experiments).
In the sentence starting from line 273, the authors claim, "Of note, the 3TC treatment was sufficient to cause a decrease in the number of neurons with hyperphosphorylated tau in the cortex and the hippocampus." However, since NFT-like inclusions did not decrease, it should be stated that the number of diffuse-type AT8-positive structures decreased.
That has been done. The new sentence is: “Of note, the 3TC treatment was sufficient to cause a decrease in the number of diffuse-type AT8-positive structures in the cortex and the hippocampus."

Reviewer 2 Report
Here, these workers report their findings on the protective effects of the reverse transcriptase inhibitor lamivudine (3TC), licensed as treatment for HIV, in P301S mice, an animal model of FTD-P associated with tau overexpression. The transgenic P301S mice were given 3TC 100mg/kg daily in drinking water which led to a decrease in. intracellular P-tau but not neurofirillary tangles (NFTs). Microglial and astrocyte activation were also reduced along with IL1b and GFAP levels. 3TC also reduced neuronal death and hippocampal atrophy in the P301S mice and prevented ventricular dilation. 3TC administration had no deleterious effects on wild type mice. Behaviourally, 3TC treatment attenuated locomotor deficits and improved short-term memory of the P301S mice. In HeLa cells co-transfected with a plasmid containing a complete LINE-1 sequence and a plasmid containing the tau sequence the LINE-1 insertion increased considerably via retrotransposon transcription compared to transfection without tau plasmids. 3TC blocked inhibited the insertion of LINE-1 sequences in HeLa DNA. The authors conclude that that the progression of tauopathy in P301S mice can be attenuated by the administration of 3TC when their first symptoms first appear.
This is a novel study aimed to determine whether the already licensed reverse transcriptase inhibitor lamivudine (3TC) could be repurposed as a protective agent in tauopathies. There are, however, a few issues that need to be considered:
First, the dose of 3TC administered here – 100mg/kg – is far greater than the 300mg daily dose administered to human HIV carriers and would be highly toxic. Have these workers considered trialling a clinical dose of 3TC in their P301S mice?
Second, it appears that 3TC clears P-tau from nerve cells but not NFTs so would have to be given to Alzheimer cases at the prodromal disease stage ahead of any tau tangle formation.
Third, the mouse model used here was of FTD-P rather than a model for Alzheimer’s disease. It is unclear why the P301S rather than the 3xTg AD mouse was chosen for study if repurposing 3TC for protective therapy in AD was the ultimate aim. While FTD can be caused by tauopathy, most cases are associated with TDP-43 aggregation.
In summary the report is of interest and has been well performed. Given the positive findings it would be interesting to repeat the study with a much lower lamivudine dose and using the 3xTg AD mouse which is more relevant to human dementia.
Author Response
This is a novel study aimed to determine whether the already licensed reverse transcriptase inhibitor lamivudine (3TC) could be repurposed as a protective agent in tauopathies. There are, however, a few issues that need to be considered:
First, the dose of 3TC administered here – 100mg/kg – is far greater than the 300mg daily dose administered to human HIV carriers and would be highly toxic. Have these workers considered trialling a clinical dose of 3TC in their P301S mice?
We do agree with the reviewer with respect to 3TC doses used in this study and in humans. We have used the same concentration of lamivudine that reference 23. In fact, we do not observe any adverse effect, on the contrary, we observe an attenuation of motor and memory deficits in tau transgenic mice and an increased life expectancy.
Second, it appears that 3TC clears P-tau from nerve cells but not NFTs so would have to be given to Alzheimer cases at the prodromal disease stage ahead of any tau tangle formation.
We have rewritten the following sentence to add reviewer suggestion:
One of the main conclusions of our work is that 3TC prevents tau pathology but not when NFTs have been already formed suggesting that 3TC should be given to Alzheimer's patients in the prodromal stage of the disease before the formation of any tau tangles. However, there is no direct evidence showing that aggregated tau is toxic for neurons. In fact, the formation of aggregates has been proposed to be a defense mechanism in other neurodegenerative diseases (i.e., Huntington's disease) [48].
Third, the mouse model used here was of FTD-P rather than a model for Alzheimer’s disease. It is unclear why the P301S rather than the 3xTg AD mouse was chosen for study if repurposing 3TC for protective therapy in AD was the ultimate aim. While FTD can be caused by tauopathy, most cases are associated with TDP-43 aggregation. In summary the report is of interest and has been well performed. Given the positive findings it would be interesting to repeat the study with a much lower lamivudine dose and using the 3xTg AD mouse which is more relevant to human dementia.
We do appreciate the reviewer recommendations. We have explored whether the retrotranscriptase inhibitor 3TC prevents tau pathology in a tauopathy mouse model based on the overexpression of human tau with a mutation present in patients with frontotemporal dementia with parkinsonism-17 (FTDP-17). We agree with the reviewer that a logical continuation of our study is to extend it to analyze the effect that this drug has on a model of Alzheimer's disease that includes the presence of senile plaques and neurofibrillary tangles as 3xTgAD model. However, we believe that this analysis goes beyond the limits of this article.